# Testing the Effects of a Virtual Reality Game for Aggressive Impulse Management: A Preliminary Randomized Controlled Trial among Forensic Psychiatric Outpatients

**DOI:** 10.3390/brainsci11111484

**Published:** 2021-11-10

**Authors:** Danique Smeijers, Erik H. Bulten, Robbert-Jan Verkes, Sander L. Koole

**Affiliations:** 1Behavioural Science Institute, Radboud University, 6525 GD Nijmegen, The Netherlands; e.bulten@pompestichting.nl; 2Forensic Psychiatric Centre Pompestichting, 6532 CN Nijmegen, The Netherlands; r.j.verkes@pompestichting.nl; 3Department of Psychiatry, Radboud University Medical Centre, 6525 GC Nijmegen, The Netherlands; 4Donders Institute for Brain, Cognition and Behaviour, Radboud University, 6500 GL Nijmegen, The Netherlands; 5Department of Clinical Psychology, VU University, 1081 HV Amsterdam, The Netherlands; s.l.koole@vu.nl

**Keywords:** aggressive behavior, intervention, VR, motivational modification

## Abstract

Prior laboratory experiments among healthy samples found that training avoidance movements to angry faces may lower anger and aggression, especially people high in trait anger. To enrich this training and make it more suitable for clinical applications, the present researchers developed it into a Virtual Reality Game for Aggressive Impulse Management (VR-GAIME). The current study examined the effects of this training in a randomized controlled trial among forensic psychiatric outpatients with aggression regulation problems (N = 30). In addition to the aggression replacement training, patients played either the VR-GAIME or a control game. Aggressive behavior was measured pre-, half-way, and post-treatment via self-report and clinicians ratings. No difference was found between the VR-GAIME and the control game. However, the participants reported gaining more insight into their own behavior and that of others. Future VR intervention tools in clinical settings may capitalize more on their benefits for self-reflection within interpersonal settings.

## 1. Introduction

Anger is an acute emotional-physiological reaction that ranges from mild irritation to intense fury and rage (i.e., state anger). The disposition to experience state anger with greater frequency and intensity is referred to as trait anger [1,2]. When anger is not controlled or regulated appropriately, it increases the risk of aggressive behavior [3,4,5]. Aggressive behavior is defined as any behavior directed to another person, object, or animal with the intention to cause harm and can be divided into in an impulsive and a deliberate subtype [6,7]. Episodes of increased anger and aggression are common to all individuals; however, when such episodes occur frequently and increase in severity, they can create substantial personal and social problems [8].

Anger regulation difficulties might increase the likelihood of aggression by increasing negative affect and physiological arousal, by reducing aggression inhibitions, compromising decision making processes, increasing difficulties in resolving difficult situations, and by diminishing the quality of interpersonal relationships (for a review, see [9]). Anger regulation problems may, therefore, be an important underlying contributing factor to engage in aggressive acts. This especially might account for a specific subtype of aggressive behavior: reactive aggression. Reactive aggression is defined as impulsive, angry or defensive responses to threat, frustration or provocation [10]. Importantly, reactive aggression is also referred to as affective aggression, i.e., spontaneous and emotionally driven forms of aggressive behavior [11]. Especially in these cases, individuals experience difficulties in regulating aggressive impulses.

In treating aggression regulation problems, interventions based on principles of Cognitive-Behavioral Therapy (CBT) have traditionally been the interventions of first choice [12,13]. CBT-based interventions, however, are only partially successful in reducing aggressive behavior and usually only beneficial to a subgroup of individuals [12,13,14,15]. One limitation of CBT-based interventions is that they appeal to abilities for self-reflection and willingness to genuinely talk about problems in controlling anger and/or aggression. However, both these abilities and this willingness are often low among individuals with anger or aggression regulation problems (e.g., [16,17]). Additionally, CBT-based interventions target more conscious, deliberate responses and thus may have little impact on underlying implicit or automatic characteristics [18]. When left untreated, these underlying factors might re-emerge during highly provoking and frustrating situations [19]. As such, there is room for alternative approaches to aggression regulation.

One such alternative approach focuses on the motivational underpinnings of anger regulation [20]. People high (versus low) in trait anger tend to have high approach motivation [21] especially in situations when they are socially provoked [22]. Approach motivation is defined as the impulse to go toward [23]. This tendency to approach potential social threats may be an important driver of heightened hostility and aggression. It even has been suggested that approach motivation determines whether trait anger becomes translated into state anger and aggression [24]. The results of multiple studies showed that higher levels of trait anger predicted more aggression when a high approach-oriented posture (leaning forward) was assumed [24]. Moreover, conditions that hinder approach behavior—like leaning backward and ambient darkness—have been found to lower state anger and aggression among people with high (rather than low) trait anger [25]. The latter findings suggests that changing people’s motivational orientation may contribute to the regulation of anger and aggression regulation among high-risk populations.

A recent set of laboratory experiments examined whether the motivational approach to anger management can be turned into a training [26]. These experiments made use of an adapted joystick task that was validated in previous motivational intervention research in the domains of treating alcohol abuse [27,28] and social anxiety problems [29]. The latter studies already proved that such an approach bias modification was successful in reducing alcohol consumption in heavy drinkers and emotional vulnerability in socially anxious individuals. In the experiments by [26], healthy participants were asked to perform a task in which they responded to angry or happy faces with a joystick. In the avoidance training condition, participants made avoidance movements to angry faces. In the control condition, participants made approach movements to angry faces. The results showed that after avoidance training, participants reported less angry feelings and expressed less aggressive impulses. The latter was most evident among individuals high in trait anger. These results suggest that reducing approach motivation towards social threatening stimuli could be an important addition to conventional anger and aggression regulation interventions.

Although the aforementioned findings are promising, they are limited in important ways. First, the avoidance training of [26] consisted of a single session. To increase the long-term effects of the training, its effects should be examined across multiple sessions. Second, the avoidance training was investigated among healthy undergraduate students who were not characterized by severe levels of anger and/or aggressive behavior. To elucidate the clinical relevance of motivational training, its effects among clinical populations need to be investigated. Third and last, the avoidance training of [26] used a joystick task that was not very engaging for participants, which could hamper implementation in clinical settings. Especially among individuals seeking treatment for anger and aggression regulation problems, treatment motivation is often lacking [16,30]. To warrant sufficient treatment motivation, it would be desirable to develop a more engaging variant of the motivational training.

To increase treatment motivation, interventions that use serious gaming and virtual reality technology have been found to be highly effective [31]. Serious games refer to games that, although fun and engaging, have training, education, or health improvement as their primary purpose [32,33]. By introducing playful and interactive elements in an intervention, serious gaming may enhance the motivation of the target group [31]. Virtual reality (VR), on the other hand, makes use of virtual environments to present digitally recreated real world activities to participants via non-immersive and immersive mediums which can be systematically manipulated to be relevant to patients’ problems [34,35]. Moreover, within VR, participants are fully immersed in the virtual environment which often creates a sense of presence. The latter refers to the participants experience of the sensation of being elsewhere, i.e., the feeling of actually being psychically present in the (VR) environment [36]. This sense of presence is affected by a variety of factors, such as interactions with avatars (i.e., VR characters). The advantages, therefore, are that VR gives the unique opportunity to investigate and treat underlying behavioral mechanisms in controlled experimental designs that nonetheless possess high ecological validity. Another advantage is that VR has the ability to induce emotions, such as higher levels of anger after provocative scenario’s [37]. Taken these advantages together, it has been suggested that VR has the potential to improve psychiatric interventions (for a review, see [38]).

Initial studies of serious gaming and VR in psychiatric treatments have found that these techniques can be used to successfully reduce aggressive behavior, impulsivity, anxiety, and posttraumatic stress symptoms and to improve self-regulation and pro-social behavior [39,40,41,42,43]. Serious gaming and VR are thus promising tools for enhancing psychological interventions and have also gained recognition in forensic psychiatry and criminology (e.g., [44,45,46,47]).

Recently, a VR aggression prevention therapy (VRAPT) was developed and examined among forensic psychiatric inpatients [48]. This intervention consisted of 16 one-hour individual treatment sessions twice a week. Their results showed that aggressive behavior did not decrease after VRAPT as compared to waiting list. However, hostility, anger control, and non-planning impulsiveness did improve after treatment, but these improvements were not maintained in a 3-month follow-up. The results of this study highlight the challenge of developing an effective VR intervention but also show that VR has potential as an intervention-tool in forensic clinical practice. However, whether the combination of serious gaming and VR can be used to train underlying automatic processes, such as an approach tendency towards a potential social threat has not yet been investigated.

For the current study, the motivational modification paradigm [26], serious gaming, and VR technology were combined to create a new treatment tool for the treatment of aggressive behavior: the Virtual Reality Game for Aggression Impulsive Management (VR-GAIME) [49]. The VR-GAIME adopted the rationale of the approach-avoidance bias modification paradigm that was investigated by [22]. Instead of joystick movements, however, the VR-GAIME manipulated whole body movements in an immersive environment. During the VR-GAIME, each participant got assigned to the role of a courier who had to collect packages in a shopping street. In the shopping street, the participant was met by avatars who were acting in either an agreeable or disagreeable manner. Patients in the experimental training condition were trained to respond with avoidance behavior to anger-relevant situations. In the control condition, patients played the same game as patients in the experimental condition but did not encounter any disagreeable avatars and hence did not receive any training about anger-relevant situations.

The aim of the current randomized controlled trial was to investigate the effect of the VR-GAIME on the level of aggressive behavior of forensic psychiatric outpatients, who were randomly allocated to the VR-GAIME or control game. In both conditions, the game was provided in combination with treatment as usual which consisted of the Aggression Replacement Training (ART) [50,51]. The original ART consists of three modules: (1) social skills training, which focuses on responding in a pro-social way to difficult situations instead of using aggression; (2) anger control training, which teaches to gain more control over aggressive thoughts and aggressive impulses; and (3) moral reasoning training, where patients learn to recognize certain cognitive distortions relating to aggression by themselves and think in a less egocentric way by means of group discussions.

Anger and aggressive impulses were measured using self-report and a validated laboratory paradigm as well as clinician ratings. Moreover, approach and avoidance tendencies were assessed using self-report. Additionally, drop-out rates among outpatients receiving aggression treatment are high [52] and are associated with psychopathy and proactive aggression [53,54]. To determine whether the drop-out number was in line with previous studies or might be lower/higher due to the game, we included measures of psychopathy and aggression subtype. Subsequently, aggression treatment might also change other emotions than anger, and the effects might affect other biases in processing facial expression. To examine this possibility, a self-report measure for distinct emotions and a measure for a hostile interpretation bias were included. We hypothesized that the combination of the VR-GAIME and ART would be more successful in reducing anger and aggressive behavior relative to the control condition. Finally, individual differences in treatment effects were explored by examining whether aggression tendency and aggression subtype, approach/avoidance tendencies, psychopathy, and emotion experience at baseline were associated with the change in aggression during treatment.

## 2. Materials and Methods

### 2.1. Design

The design was a double blind randomized controlled trial. The trial had two conditions: (1) ART and the VR-GAIME and (2) ART and the VR control game. Assessments took place pre-, halfway, and post-treatment. The sample size was calculated for the main research question using G*Power software. The sample size was calculated for a 2 (group: ART and VR-GAIME vs. ART and placebo game) × 3 (assessment: pre vs. halfway vs. post) interaction, with the assumption of a small-to-medium effect size (eta2 = 0.08) and a power of 1-β = 0.80. This led to a minimum required sample size of 60.

### 2.2. Participants

From 1 January 2018 to 1 June 2019, 305 forensic psychiatric outpatients were referred to “Kairos”, the outpatient unit of the Forensic Psychiatric Clinic the Pompe Foundation in Nijmegen, The Netherlands, because of aggression regulation problems. Admission to Kairos occurs on either obligatory (when sentenced by a judge) or voluntary basis (based on reference by general practitioner which is necessary in secondary care). Inclusion in the study required meeting the following criteria: (1) male sex and (2) aggression regulation treatment was indicated. The antisocial and borderline personality disorders as well as the intermittent explosive disorder were the most common psychopathologies. However, the current study and the aggression regulation treatment had a transdiagnostic focus. No distinction was made based on specific psychopathology. Patients were excluded from the study if they met the following exclusion criteria: (1) current major depression, (2) current severe addiction, (3) lifetime bipolar disorder, (4) lifetime psychosis. These patients were excluded because under such conditions, a proper treatment of these disorders will be a priority, and furthermore, these conditions will seriously limit the responsivity of patients to aggression regulation treatment. This procedure is in line with previous studies in the same population, e.g., [54]. In the current study, 51 (of the initial 305) male forensic psychiatric outpatients were eligible and willing to participate. An overview of reasons for exclusion of the remaining 254 patients is provided in Table 1.

Of the 51 forensic psychiatric outpatients, five were excluded because their treatment was postponed until after the inclusion period of the study. Pre-treatment measurements were collected from 46 forensic psychiatric outpatients. Sixteen patients dropped-out of the current study because of cybersickness (N = 1), no show during intervention (N = 10), reference to other type of treatment (N = 1), not willing to participate anymore (N = 3), injury due to which playing the game was not possible anymore (N = 1). Unfortunately, no extra follow-up information regarding this drop-out group is available. The remaining 30 patients performed both the half-way and post-treatment measurements. Demographic information is provided in Table 2.

The present study was reviewed by the Medical Ethical Committee (METc) of the VU medical center. The committee judged that the Medical Research Involving Human Subjects Act (WMO) did not apply to this study (2017.563). Moreover, the study was approved by the local ethics committee of the Faculty of Social Sciences at the Radboud University in Nijmegen (ECSW2017−1303−499). The trial was registered with The Netherlands National Trial Register, number: NTR6986.

### 2.3. Procedure

Clinicians at Kairos asked patients who were referred to aggression regulation treatment (group or individual), whether they agreed to be contacted about the study. When they agreed, patients were contacted by the researcher. All patients received treatment as indicated whether they participated in the study or not. After receiving information about the nature of the study, forensic psychiatric outpatients were asked to assign a consent form. Subsequently, they were screened by the researchers (who were also trained clinicians in the use of these interviews) with the Structured Clinical Interview for DSM-IV axis II personality disorders (SCID-II) [56], the Research Criteria set for Intermittent Explosive Disorder (IED-IR) [57], and the MINI International Neuropsychiatric Interview for axis I disorders (MINI) [58,59] regarding the aforementioned exclusion criteria and in order to confirm diagnosis. Once patients were found suitable for participation, they proceeded with the pre-treatment measurement.

The pre-treatment measurement consisted of several questionnaires and two computer tasks. After this assessment, participants started with their treatment. The forensic psychiatric outpatients were assigned by a computerized random number generator to one of two conditions: (1) ART and VR-GAIME or (2) ART and VR control game. In both conditions, patients were asked to play the game at the outpatient clinic alongside the first five sessions of their treatment. After five weeks, after the last VR session, the level of aggressive behavior was determined by use of a questionnaire. The post-treatment measurement took place after 12 weeks and consisted of the same questionnaires and computer tasks as the pre-treatment measurement. After the post-treatment measurement, patients were informed about which condition they participated in and a manipulation check was conducted. The difference between the two game versions was explained. In case patients participated in the control condition, the opportunity was offered to play the experimental game. Patients were also asked to indicate how much they enjoyed the VR-GAIME and whether they thought it was of added value on their treatment on a scale from 1 to 5. During a brief qualitative interview after the post-treatment measurement, patients were asked to elaborate on their experience with playing the VR-GAIME and on why/why not they thought the VR-GAIME was of added value (e.g., what they have had learned in addition to treatment as usual or what should have been included to increase the added value). Patients were compensated for their participation with a monetary reward.

### 2.4. Intervention

#### 2.4.1. Treatment as Usual

All forensic psychiatric outpatients who were referred to Kairos because of aggression regulation problems received the ART [50,51]. The ART is a CBT based intervention. Besides ART for general aggression and violence, ART is also offered for perpetrators of intimate partner violence. This version of the ART is identical to the regular ART except that the partners of the patients were involved during this intervention (N = 7). Both the regular ART as well as the ART for domestic violence perpetrators consisted, as offered by Kairos, of two of the three original modules: (1) social skills training and (2) anger control training. An additional module of psychomotor therapy was included to improve anger control. Both interventions occurred either in groups (N = 24) or individually (N = 7) and consisted of two 90 min weekly sessions during 12 weeks. The first 10 weeks consisted of the social skills and anger control training. Week 11 consisted of a session to integrate all that was learned in the previous weeks. Finally, week 12 consisted of an evaluation session. Indication for ART is determined by a multidisciplinary team. The ART therapists (all clinicians at “Kairos”, not involved in the current study as a researcher) were all formerly trained in applying the ART and, in addition, made use of a detailed intervention manual and participated in intervision.

#### 2.4.2. VR-GAIME

Before the VR-GAIME was developed, interviews with clinicians from the forensic psychiatric in- and outpatient units of the Forensic Psychiatric Clinic the Pompe Foundation in Nijmegen, The Netherlands, were conducted to investigate which kinds of computerized anger and/or aggression regulation tools were available and what the needs of the clinicians were. They had worked with a mobile app before which enabled patients to gain insight into their personal risk factors. Clinicians experienced this tool as hard to implement, and it did not receive positive patient evaluations. A computerized intervention the clinicians were extremely enthusiastic about was the aforementioned VRAPT intervention [48]. Clinicians believed in the added value of VR because it motivated patients to receive treatment and to explore and practice skills they had learned during treatment. Based on these clinical experiences, it was decided to choose VR technology for the current treatment tool.

The virtual environment was created by CleVR BV (Delft, The Netherlands). With permission of [48], the shopping street environment was used as basis for the VR-GAIME. VRAPT consists of an interactive role play; the virtual environments and avatars were controlled by the VRAPT therapist. However, the VR game needed to be more standardized to be able to train specific avoidance behavior towards potential social threats and to use it repeatedly. Therefore, it was decided that the player of the game followed a predetermined route during which he encountered avatars. Furthermore, the VR game was based on the underlying principles from the training developed by [26]. Only arm movements to train avoidance behavior in an immersive environment seemed rather limited. Based on extensive studies on the effect of approach/avoidance-oriented body postures [24], instructing players to lean forward (approach) or backward (avoidance) was chosen as alternative. Furthermore, following the philosophy of a ‘serious game’ [31], the training was developed to be fun and challenging.

Forensic psychiatric outpatients received written instructions of what the game entailed as well as a demonstration of all actions during the game. During the game, patients wore an Oculus Rift 2, a head-mounted display. The virtual environment moved automatically as if the patient was actually walking down a shopping street. Patients were able to walk themselves within pre-determined boundaries. To make sure patients could play the game safely, a guardian system was set up. Due to this guardian system, a virtual laser cage was displayed once a patient cannot walk further in that direction in the real world. This virtual cage ensured that the patient cannot bump into objects in the surroundings. To warrant patients’ safety, the researcher was present in the room at all times and gave instructions to reset the patients position if needed.

The VR game had five levels, which were ascending in level of difficulty. During the game, the patient was working as a mail courier. This scenario was chosen because it fitted well within the shopping street environment. Both the environment and the scenario were invented by CleVR’s BV game and concept developers. The back-story was that while the courier was driving, he lost several packages. In each level, the patient had to walk down a shopping street in order to collect the lost packages. Once enough packages were collected, the patient could proceed to the next level. Figure 1 shows the home screen of the game and the shopping street the patients virtually walk around. In the shopping street, the patient was met by avatars who act in either an agreeable or disagreeable manner (see Figure 1). The behavior of the avatars was experimentally manipulated to provide the training component of the game. Agreeable avatars displayed neutral or happy facial expressions and said pleasant things (e.g., “nice to see you”, “have a good day”, “you are wearing such a nice T-shirt”), whereas disagreeable avatars displayed angry facial expressions and said less pleasant things (e.g., “what are you looking at”, “get out of my way”, “what do you want from me”). Patients received the instruction that it was important to respond correctly towards the avatars. In the experimental condition, the instruction was to lean forward (i.e., make an approach movement) in response to agreeable avatars and to lean backwards (i.e., make an avoidance movement) in response to disagreeable avatars. In each level, four agreeable and four disagreeable avatars appeared. This resulted in 20 approach and 20 avoidance movements per session. When an incorrect response was given towards disagreeable avatars, the patients lost a package.

Instead of the active (experimental) game, half of the patients played a control game. In terms of the general game elements, the control game was identical to the original game. The only difference was that, in the control game, no disagreeable avatars appeared. As a consequence, avoidance behavior was not trained in the control condition. This resulted in 40 approach movements per session. The instruction patients received was comparable. This setup was chosen to avoid that patients easily could find out in which condition they participated. Based on prior studies, we know that patients tend to discuss such experiences once they participate in group treatment. Patients were randomly allocated to an experimental or a control condition. Both versions of the game had a maximum duration of 30 min. This length was chosen to keep the burden for patients to a minimum but still provide enough training trials. This approach is in line with other studies on approach-avoidance bias modification in combination with treatment as usual (e.g., [28,60]). The two versions of the game were referred to as game 1 and game 2 on the VR computer. Furthermore, the researcher took position behind the computer screen in order to stay blind for the condition.

Besides the packages and the avatars in the shopping street, the patient also came across litter on the street. The patient had to make sure he did not walk into this litter otherwise he would lose a package. Thus, when confronted with litter, the patient had to pick it up and deposit it behind him. The latter game element was introduced to make the game more varied and engaging. An additional game element consisted of the mini-game. This was an extra challenge and was included to make the game more varied for players. Patients gained entry to the mini-game by collecting bonus packages, which appeared as more colorful and somewhat smaller as the regular packages. Once enough bonus packages were collected, the mini-game could be played. During the mini-game, the patients stood underneath a window from which packages were thrown down. Patients needed to catch these fallen packages. The higher the level, the more difficult the mini-game was: the packages fell faster, and some rubbish was thrown out of the window which should not be caught. All game elements were included to induce the sense of presence, which is considered important for VR interventions [36].

### 2.5. Measures

#### 2.5.1. Questionnaires

The Social Dysfunction and Aggression Scale (SDAS) [61] is an observer scale that measures the severity of state aggressive behavior. It consists of nine items measuring aggression directed to others and two items measuring aggression directed to the self. Items have to be scored on a 4-point scale with 0 = not present and 4 = severely to extremely present as extremes. In the current study, the SDAS was rated by the clinician as well as the patient himself. In both bases, aggressive behavior was rated over a period of two weeks. The SDAS was administered three times: pre-, halfway, and post-treatment. The SDAS as self-report demonstrated acceptable test–retest stability and internal consistency in prior research; intraclass correlation coefficients ranged from 0.65 to 0.82 and Cronbach’s α from 0.76 to 0.82 [54]. In the present study, the internal consistency was also proven to be good as self-report (Cronbach’s α pre-treatment = 0.87; half-way = 0.84; post-treatment = 0.85) and as observer scale (Cronbach’s α pre-treatment = 0.83; half-way = 0.84; post-treatment = 0.89).

The Behavioral Inhibition System/Behavioral Activation System (BIS/BAS) [62] scale is a well-validated scale, containing 20 items measuring individual differences in personality dimensions that reflect the sensitivity of two motivational systems: the aversive (BAS) and the appetitive (BIS) system. The BIS subscale consists of 7 items whereas the BAS subscale consists of 13 items. The BAS subscale is subdivided in three subscales: fun seeking (4 items), reward responsiveness (5 items), and drive (4 items). Participants rate how true the statements are for them on a 4-point Likert scale (0 = very true, 4 = very false). Prior research has shown that the Dutch translation exhibits adequate internal consistency (Cronbach’s α ranging from 0.59 to 0.79) [63]. In the present study, the internal consistencies ranged from poor to acceptable (Cronbach’s α BIS = 0.59; BAS fun seeking = 0.29; BAS reward responsiveness = 0.56; BAS drive = 0.71). The BIS/BAS was administered at pre- and post-treatment.

The Reactive Proactive Questionnaire (RPQ) [64,65] is a 23-item self-report questionnaire to measure reactive and proactive aggression. The reactive subscale consists of 11 items whereas the proactive subscale consists of 12 items. The items are rated 0 (never), 1 (sometimes), or 2 (often). Prior research indicates that the Dutch translation has good internal consistency (Cronbach’s Alpha = 0.91) and adequate convergent (all r < 0.16), criterion (delinquents from prison and forensic mental health scored higher than non-offenders), and construct validity (violent offenders show more proactive aggression than non-offenders, *p* < 0.001) [64]. In the current study, the internal consistency was good (Cronbach’s α = 0.81 for reactive aggression and 0.85 for proactive). The RPQ was administered at pre- and post-treatment.

The Aggression Questionnaire (AQ) [66] is a self-report questionnaire to assess an overall trait of aggression. It consists of 29 items that are divided into four subscales: physical aggression, verbal aggression, anger and hostility. The items are scored on a 5-point Likert scale (1 = extremely unlike me to 5 = extremely like me). Prior research indicates that the Dutch translation has adequate psychometric properties (Cronbach’s α = 0.86) [67]. In the present study, the internal consistency ranged from acceptable to good (Cronbach’s α ranging from 0.68 to 0.85). The AQ was administered pre- and post-treatment.

The Self-Report Psychopathy Short-Form (SRP-SF) [68] is a self-report measure of adult psychopathic features. The SRP-SF consists of 29 statements that are divided into four subscales: Interpersonal manipulation, callous affect, erratic life styles, and criminal tendencies. Participants have to rate the extent to which they agree with these statements on a 5-point Likert scale (1= disagree strongly, 5 = agree strongly). Prior research indicates that the Dutch version of the SRP-SF has good internal-consistency and test–retest reliability (Cronbach’s α ranging from 0.58 to 0.73 and r ranging from 0.60 to 0.86) [69]. In the current study, the internal consistency ranged from questionable to good (Cronbach’s α ranging from 0.66 to 0.84). The SRP-SF was only administered pre-treatment.

The State Trait Anger Scale (STAS) [70] has been designed to measure state and trait anger. It is a self-report questionnaire of 20 items subdivided in two subscales: state and trait anger. State anger refers to an emotional condition of a patient, which is consciously experienced and fluctuates over time. Trait anger refers to a stable personality quality: the disposition to become angry, a tendency that differs much among individuals. Prior research indicates that the Dutch translation has proven psychometric properties [71]. In the present study, the internal consistency was also excellent (Cronbach’s α = 0.95 for trait anger to 0.97 for state anger). The STAS was administered pre- and post-treatment.

The Discrete Emotions Questionnaire (DEQ) [72] is a self-report measure of the following distinct state emotions: fear, anxiety, sadness, anger, disgust, happiness, relaxation, and desire. The participants are asked to think of someone with whom they have many conflicts. Subsequently, they are asked to indicate to what extent they experience these emotions regarding this person on a 7-point Likert scale (1 = not at 7 = an extreme amount). Thirty-six emotions are listed which are all synonyms for the eight aforementioned emotions. The original version has good internal consistency (Cronbach’s Alpha ranging from 0.95 to 0.97) [72]. In the current study, the internal consistency was acceptable to good (Cronbach’s α ranging = 0.71 pre-treatment; 0.84 half-way; 0.87 post-treatment). The DEQ was administered pre-treatment, half-way, and post-treatment in order to assess changes in state emotions during the intervention.

#### 2.5.2. Paradigms

The virtual voodoo doll task (VVDT) [26] was used to measure aggressive impulses. In this task, a picture of a doll is presented on the computer screen. Participants are asked to think about a person who did them wrong. Subsequently, they receive the following information: “In previous research, we have found that you can get rid of negative energy by taking action in response to the person that caused you harm. Imagine that the virtual doll is the person from the situation you just recalled. You now have the opportunity to insert as many pins in the doll at any location you like.”

Participants are able to pick up a pin-presented next to the doll by left-clicking with their mouse, move it around the screen, and then push the pin into the doll at any place they want. Once a pin was pushed into the doll, it looked as though the pin was stuck in the doll. The position of the pin can be changed as desired. A maximum of 50 pins can be inserted into the doll. The total number of pins each participant will put into the doll was registered. Furthermore, a screen-capture of the doll was saved in order to code the position of the pins. The traditional voodoo doll task, with an actual doll and pins, is a reliable measure with convergent and construct validity [73]. Moreover, the virtual version of the voodoo doll task is thought to differentiate well between different levels of aggressive impulses, as indicated by putting pins in the vital or less vital body parts (Cronbach’s α ranging from 0.88 to 0.91) [26]. The VVDT will be administered pre- and post-treatment.

The Hostile Interpretation Bias Task (HIBT) [74] was used to assess a HIB at pre- and post-treatment measurement. Photos of faces with emotional affect (angry, fearful, disgusted, happy) of four male and four female models were selected from the Radboud Faces Database [75]. Each affective picture was morphed (using WinMorph 3.01) five times with the neutral image of the same individual, creating 20%, 40%, 60%, 80%, and 100% emotion intensity, respectively. The neutral expression was in all models displayed with mouth closed whereas the emotional pictures had their mouth open. This difference in mouth-opening resulted in pictures showing ambiguous expressions. The task consisted of a practice block and two experimental blocks. The practice block consisted of 16 trials (8 models × 2 emotions). Only pictures with happy and angry affect and of 100% intensity were used to familiarize participants with the task. Each experimental block consisted of 168 trials (8 models × 4 emotions × 5 intensity levels + 8 neutral images). The order of the pictures is pseudo-randomized and equal in both blocks.

Participants were instructed to indicate whether the picture looked hostile or not. When participants believed they saw a hostile picture, they had to press the Z-key, otherwise the M-key (on a qwerty keyboard). Participants had to respond as quickly as possible. The picture, size 8.5 cm × 10.5 cm, was presented for four seconds, on the center of the computer screen, against a black background. The pictures remained on screen until a response was given or until four seconds had passed. After a pretrial pause of one second, a new picture was displayed immediately. Labels were displayed on the left (Yes, hostile) and right (No, not hostile) bottom corner of the screen in white Arial font, size 30. Responses given by pressing the Z-key, indicating that the participant saw a hostile picture, are defined as “hostile” responses. If a response was not given within four seconds, the words “Too late” appear on the screen in red. A hostile interpretation bias was defined as the percentage of “hostile” responses to the emotional pictures. The hostile responses were dummy coded (0 = no, not hostile, 1 = yes, hostile), and the mean was calculated which indexed the percentage of the pictures that were interpreted as hostile. Trials without a response (due to late responding) were not taken into account. The HIBT was shown to have good test-retest reliability in prior research (r ranging from 0.295 to 0.908) [74].

### 2.6. Statistical Analyses

First, a Multivariate Analysis of Variance (MANOVA) was conducted to investigate whether forensic psychiatric outpatients in the two conditions, patients who received regular ART vs. ART for domestic violence perpetrators, and patients who received group versus individual treatment differed regarding age and IQ and regarding the questionnaires and paradigms at pre-treatment measurement. Subsequently, to examine whether aggressive behavior changed over time and whether this change was different across condition, we used a linear mixed model (SPSS, version 25). One advantage of this analysis is that it is possible to include individuals with incomplete data, without imputing data [76]. This method was favored because the halfway and/or post-treatment measurements were not completed by all participants and/or clinicians.

Main effects of training. The basic model was a repeated-measures design with aggressive behavior as measured with the self-report SDAS as dependent variable, Time of measurement (pre-, half-way, post-treatment) as within-subjects factor, and Condition (VR-GAIME vs. control) as between-subjects factor. Repeated covariance type was set at diagonal, which assumes heterogeneous variances and no correlation between elements [76]. With respect to Time, the slope was set as a fixed effect and the intercept as a random effect. This random effect was defined in order to assess variation in the dependent variable because variation among individuals, regarding change in aggression over time, was assumed [77,78]. The covariance type for the random effects was set at unstructured as a completely general covariance matrix [76].

Second, a similar linear mixed model was conducted, now with the SDAS rated by the clinician as dependent variable and Time of measurement (pre-, half-way, post-treatment) as within-subjects factor and Condition (VR-GAIME vs. control) as between-subjects factor, to examine whether aggressive behavior decreased during treatment according to clinicians. We conducted another similar linear mixed model with the DEQ as dependent variable and Time of measurement (pre-, half-way, and post-treatment) as within-subjects factor and Condition (VR-GAIME vs. control) as between-subjects factor, to examine whether distinct emotional experiences change during treatment.

Thirdly, we investigated whether the secondary outcome measures, which related to trait aggression, trait anger, reactive and proactive aggression, aggressive impulses, behavioral inhibition and disinhibition, and hostile interpretation bias, changed after treatment and whether this change was different among the two training conditions. A linear mixed model can only analyze one dependent variable. As the current sample size was relatively small, a large number of linear mixed models could not be executed. Therefore, a different statistical approach was chosen: difference scores for all secondary outcome measures were calculated; the pre-treatment score was subtracted from the post-treatment score. Negative scores indicated a reduction during treatment whereas positive scores indicated an increase during treatment. Subsequently, we conducted a MANOVA to explore whether change in secondary outcome measures during treatment differed between the two training conditions.

Individual Differences in Treatment Effects. To explore individual differences in the effects of Treatment, we analyzed as potential moderators trait aggression, trait anger, reactive and proactive aggression, aggressive impulses, psychopathy, behavioral inhibition and disinhibition, and hostile interpretation bias measured at pre-treatment. We examined these moderators by adding main effects of these variables and two-way interactions of the measures and Time/Condition and the three-way interaction between potential moderator, Time, and Condition to the basic linear mixed model, with SDAS as self-report, as described above. To be able to interpret the results, all independent variables were centered; the sample mean was subtracted from the individual participant’s mean. These analyses were only conducted for the patient-rated SDAS. This approach is in line with [54].

## 3. Results

### 3.1. Differences among Forensic Psychiatric Outpatients

Table 3 displays the means on all questionnaires, paradigms, and VR-GAIME. Means are presented for the total sample as well as separately for the VR-GAIME and control condition. A MANOVA was conducted to investigate whether patients in the two conditions, patients who received regular ART vs. ART for domestic violence perpetrators, and patients who received group versus individual treatment differed regarding age, IQ, and regarding the questionnaires and paradigms at pre-treatment measurement. No significant multivariate effect of VR-GAIME vs. control game, ART vs. ART for domestic violence, and group vs. individual treatment emerged; Wilks’ Lambda = 0.312, F(20, 7) = 0.77, *p* = 0.697, ηp^2^ = 0.688; Wilks’ Lambda = 0.058, F(21, 4) = 3.07, *p* = 0.143, ηp^2^ = 0.942; Wilks’ Lambda = 0.149, F(21, 4) = 1.09, *p* = 0.528, ηp^2^ = 0.851, respectively.

### 3.2. Main Effects of Training

In both conditions, patients correctly approached most of the agreeable avatars (see Table 3). The patients in the experimental condition also correctly avoided the majority of the disagreeable avatars. This indicates that, in both conditions, participants largely adhered to the instructions. Subsequently, a linear mixed model was conducted to examine whether aggressive behavior changed over time and whether this change was different across condition. The analysis of the basic model “Time, SDAS self-report” revealed a marginally significant main effect of Time, indicating a possible reduction in aggression during treatment (see Table 4). The analysis of the basic model “Time, SDAS clinician” again revealed a significant effect of Time, indicating that aggressive behavior reduced during treatment according to clinicians. Finally, the basic model “Time, DEQ” revealed no significant effect of Time. The interaction effects of Time × Condition did not reach significance in any of the basic models (see Table 4). As an effect, size is not provided by linear mixed models; Cohen’s D was calculated by dividing the mean difference between pre- post SDAS/DEQ scores by the pooled standard deviation: SDAS self-report: 14.04–10.60/(√ ((7.852 + 7.092)/2)) = 0.46; SDAS clinician: 12.24–9.87/(√ ((6.842 + 7.382)/2)) = 0.33; and DEQ: 3.03–2.87/(√ ((.822 + 0.932)/2)) = 0.18.

Subsequently, a MANOVA was conducted to explore whether change in secondary outcome measures during treatment was different between the two training conditions. Bonferroni correction was used to control for multiple comparisons. No significant multivariate effect of VR-GAIME vs. control game emerged; Wilks’ Lambda = 0.392, F(14, 14) = 1.55, *p* = 0.210, ηp^2^ = 0.608.

### 3.3. Exploratory Analyses; Individual Differences in Treatment Effects

The basic model (Time, SDAS self-report) was extended by adding psychopathy, trait aggression, state and trait anger, reactive and proactive aggression, aggressive impulses, behavioral inhibition and disinhibition, and hostile interpretation bias as possible predictor variables. Significant main effects of Time, trait anger, reactive aggression, and BAS fun emerged (see Table 4), suggesting that aggression changed during treatment and that these characteristics were associated with differences in aggressive behavior. Time and BAS fun were negatively associated with treatment responsivity, whereas trait anger and reactive aggression were positively associated.

In the subsequent model, 2-way and 3-way interactions were included to examine which characteristics might explain variability in aggression reduction during treatment, and whether this differed across condition. The analysis of this model revealed a significant positive main effect of trait anger and reactive aggression (see Table 4). Moreover, there was a marginal significant interaction of Time × Condition × reactive aggression (see Table 4). In the final model, non-significant interactions were removed. The main effects of trait anger and reactive aggression remained significant. Furthermore, reactive aggression, measured at pre-treatment, was marginally associated with the course of treatment and differed between conditions. High baseline scores on reactive aggression marginally predicted a more rapid decrease of aggressive behavior in the control condition but not in the experimental condition. None of the interactions with condition was significant.

### 3.4. Post-Training Questionnaire

Six patients thought they had played the VR-GAIME; 16 thought they had participated in the control condition, and nine did not have an idea. Of all 30 patients, only eight were correct about the condition in which they had participated. Furthermore, they enjoyed playing the game (M = 3.54, SD = 1.14), and they moderately thought that the VR-GAIME was of added value (M = 2.27, SD = 1.37). Patients explained that the VR environment (both conditions) gave them insight in what triggered their irritation (N = 4); due to the game, they realized that other people can be positive without ulterior motives (N = 4); they also realized that their evaluation of avatars depends on their personal mood in that particular day (N = 2); the control game calmed some patients down (N = 2) because there were only positive avatars; some patients in the experimental condition believed the VR-GAIME had an unconscious effect (N = 3); one patient explained that he had learned to first take a good look before he reacts; another patient stated that due to the game he had learned to handle crowded environments, and several patients believed the game was relatively unchallenging from level three and thought that it could be improved by adding different situations (N = 9).

## 4. Discussion

Priori laboratory studies indicated that training avoidance movements to angry faces may lower anger and aggression among people with aggression-prone personality traits [26]. Based on this basic principle, we developed a VR-GAIME and tested its efficacy in a randomized controlled trial among a group (N = 30) of forensic psychiatric outpatients characterized by aggression regulation problems. Results suggested that aggressive behavior reduced over the course of treatment. Moreover, high levels of trait anger and reactive aggression were associated with higher levels of aggressive behavior during the course of treatment. Contrary to expectations, the VR-GAIME was not more successful in reducing anger and aggressive behavior relative to the control condition. In the following paragraphs, we consider possible reasons for the lack of effectiveness of the VR-GAIME, along with ways in which future work may realize the unfulfilled potential of combining serious gaming and VR in creating effective aggressive management interventions.

We see four potential explanations why the VR-GAIME could have failed to show effects in the present research. A first potential reason for the lack of effects of the VR GAIME is theoretical: perhaps training avoidance behavior does not reduce anger and aggression among aggression-prone individuals. We regard this potential explanation as unlikely. This is because prior findings by our team were clear-cut, in that we were reliably able to demonstrate that training avoidance behavior to angry faces lowers anger and aggression among aggression-prone people [26]. A second potential explanation for the lack of effects of the VR-GAIME is lack of statistical power: our sample was twice as small as we had originally aimed for, which meant that the present research had limited statistical power. Although we grant that the present study was limited in this regard, we were still able to replicate the effects of individual differences in trait anger and aggressive personality. Because these individual differences are well-established and tend to have statistical effects in the small-to-moderate range, our individual-difference findings suggest that our study design was still sufficiently sensitive to detect small-to-moderate effects. Consequently, we are inclined believe that the lack of effects of the VR-GAIME was not, or at least not entirely, due to lack of statistical power.

A third potential explanation for the lack of effects of the VR-GAIME is the distinctive population we examined in the present study. In prior studies, the avoidance training had been used among university students. By contrast, in the present study, our sample consisted of forensic psychiatric outpatients. It is conceivable, and even plausible, that forensic psychiatric outpatients are a more difficult group to influence with any kind of training program. Moreover, it may be that forensic psychiatric outpatients are not uniformly characterized by reactive approach motivation. Theoretically, the motivational training should only be effective among people whose anger management problems derive from an excess of reactive approach motivation. In future research, it would be helpful to conduct more refined diagnostic tests to screen out those forensic psychiatric outpatients who are most likely to benefit from motivational avoidance training.

Finally, a fourth potential explanation for the lack of effects of the VR-GAIME may be that the translation of the intervention to a gamified VR-format was imperfect. We regard this explanation as plausible. The design process of the VR-GAIME was complex and far from straightforward. We had to make many design decisions (e.g., the cover-story, specific gaming elements, how different points could be earned) ‘in the blind’, without a clear theoretical rationale. As a results, it is possible that the final version of the VR-GAIME diverged so much from the original joystick training [26] that it could no longer be expected to have the same effects. It is important to note that the decisions about the specific training components were in line with previous research. An important avenue for future research on gamification of neurocognitive tasks is to carefully validate the translation of the original to the new task before implementation in research/clinical practice. Unfortunately, in the current study, the timespan for a thorough validation procedure was too limited. Moreover, important differences exist between the two conditions of the VR-GAIME that might have played a role in the unexpected findings and that need to be reconsidered in future research. In both versions, in each level, eight avatars approached the patient. This entails that in the experimental condition, each level consisted of four agreeable and four disagreeable avatars, but in the control game, each level consisted of eight agreeable avatars. The latter could have provided a general positivity training. A type of training which can have beneficial effects in its own right and is based on the idea that individuals with emotional disorders are characterized by a lack of positive biases [79].

Even though the VR-GAIME did not yield the expected reduction of aggressive behavior during treatment, the game did have another important clinical effect. Several patients reported that playing the game gave them insight in their own as well as in others behavior. These findings are in line with the recent VRAPT study among forensic psychiatric inpatients [48]. This study found no significant reductions in aggressive behavior after VRAPT compared to a waiting list but patients, as well as clinicians, were positive about this intervention. Moreover, after VRAPT, patients reported more insight into their triggers and more awareness of their physiological arousal. In general, forensic psychiatric outpatients are thought to display a lack in reflection and introspection (e.g., [16,17]). It would clinically be highly relevant if a tool/intervention such as the VR-GAIME or VRAPT could help improve these abilities. Prospective research should elucidate which component of the game increased this insight and how this can be developed in further detail in order to create an add-on tool for this specific purpose. Specific attention should be paid to the outcome measures as the materials used in the current study focused on emotional and behavioral components and did not give insight in changes in reflective abilities. Several patients also reported that they believed the game was little challenging after playing it a few times and that it did not have enough variety. The VR-GAIME may thus be developed further and expanded with different situations and possibly also with more challenging components.

The present study inevitably has limitations. First, due to difficulties with the data collection and high drop-out rates, the required sample size was not met. Consequently, the current sample size was small, and the study is underpowered. The results have to be interpreted with care and replicated in larger samples. Second, except for one, all measurements consisted of self-report. It is questionable whether a population of forensic psychiatric outpatients is fully able to reflect on their own behavior and whether they are willing to answer genuinely. Third, some measurements showed poor reliability. These results have to be interpreted with care. Fourth, no follow-up measurement was included. This would enable one to determine the long-term effects or possible delayed effects of the VR-GAIME and ART. Fifth and last, the present study included only male forensic psychiatric outpatients, which means that the current findings may not be generalizable to a female population with aggression regulation problems.

## 5. Conclusions

The results of this preliminary randomized controlled trial showed that the VR-GAIME was not more successful in reducing anger and aggressive behavior relative to the control condition in forensic psychiatric outpatients with aggression regulation problems. The qualitative data, however, point in the direction of “positive side-effects” on motivation and problem insight. We hope that the present article will promote further innovations in the treatment of anger and aggression problems. These problems are associated with tremendous personal and social costs. It therefore remains important to develop new and more effective interventions for managing anger and aggression. We continue to believe that serious gaming and VR technology still hold considerable promise in this domain. The current study hopefully provides new leads for fulfilling this promise.

## Figures and Tables

**Figure 1 brainsci-11-01484-f001:**
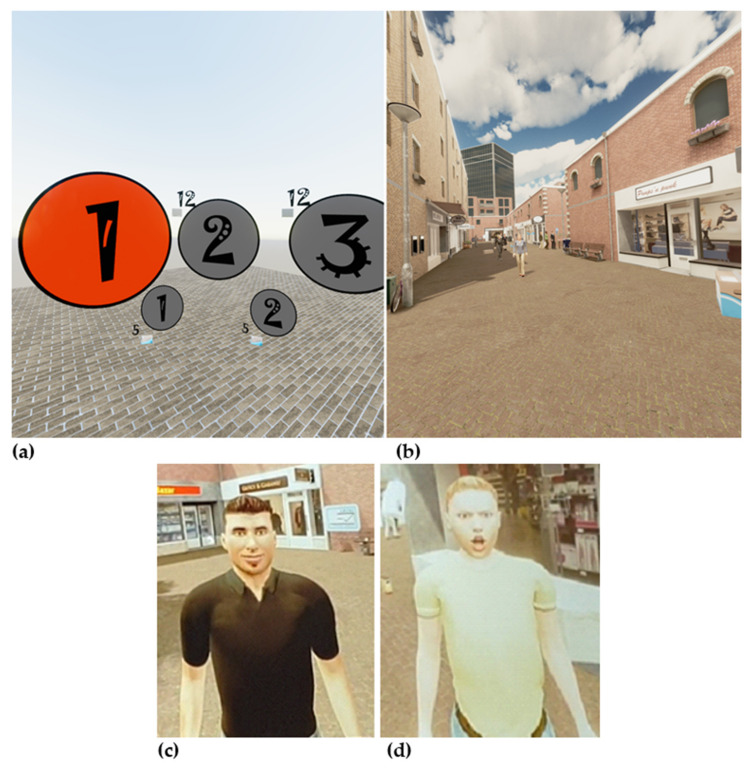
Virtual Reality Game for Aggression Impulsive Management (VR-GAIME), home screen (**a**), virtual shopping street (**b**), agreeable avatar (**c**), and disagreeable avatar (**d**).

**Table 1 brainsci-11-01484-t001:** Reasons for exclusion.

	N
Total	254
Reason:	24
Negative decision by therapist due to severity of psychopathology	69
Dropout after intake/not suitable for treatment	92
Refused to participate	
Exclusion criteria:	
- Female	10
- Current major depression	2
- Lifetime bipolar disorder	1
- Lifetime psychosis	6
- Current severe alcohol/drug dependency	2
- Insufficient understanding of Dutch language	3
- No current aggressive behavior (only past)	45

**Table 2 brainsci-11-01484-t002:** Demographic information.

	Mean/N
Age	*M* = 36.13 (*SD* = 12.88)
IQ*	*M* = 83.68 (*SD* = 14.98)
Alcohol use, unit/week	*M* = 2.06 (*SD* = 4.56)
Cannabis use, joint/week	*M* = 2.07 (*SD* = 4.09)
Obligatory admission	N = 5
Voluntary admission	N = 25
Diagnosis:	
- Antisocial personality disorder	N = 11
- Borderline personality disorder	N = 4
- Intermittent explosive disorder	N = 23
- ADHD	N = 9
- History of depressive disorder	N = 13

*as measured by using the Dutch Adult Reading Test [55].

**Table 3 brainsci-11-01484-t003:** Descriptives (Mean, SD) of the two treatment conditions and the total sample.

	VR-GAIME (N = 15)	Control (N = 16)	Total Sample (N = 31)
Questionnaires			
SDAS pre-treatment	M = 14.47 (SD = 8.55)	M = 13.25 (SD = 8.35)	M = 13.84 (SD = 8.33)
SDAS half-way	M = 14.00 (SD = 6.74)	M = 12.25 (SD = 8.35)	M = 13.07 (SD = 7.57)
SDAS post-treatment	M = 12.20 (SD = 7.13)	M = 9.00 (SD = 6.91)	M = 10.60 (SD = 7.09)
DEQ pre-treatment	M = 2.94 (SD = 0.81)	M = 3.21 (SD = 0.86)	M = 3.08 (SD = 0.83)
DEQ half-way	M = 2.69 (SD = 0.67)	M = 2.82 (SD = 0.62)	M = 2.76 (SD = 0.64)
DEQ post-treatment	M = 2.74 (SD = 0.93)	M = 3.00 (SD = 0.93)	M = 2.87 (SD = 0.92)
SRP-SF pre-treatment	M = 74.00 (SD = 11.86)	M = 69.5 (SD = 15.15)	M = 71.68 (SD = 13.63)
AQ pre-treatment	M = 96.73 (SD = 18.80)	M = 89.75 (SD = 14.52)	M = 93.13 (SD = 16.82)
AQ post-treatment	M = 90.93 (SD = 22.35)	M = 84.60 (SD = 18.13)	M = 87.77 (SD = 20.25)
RPQ reactive pre-treatment	M = 15.13 (SD = 4.21)	M = 13.88 (SD = 4.01)	M = 14.48 (SD = 4.09)
RPQ reactive post-treatment	M = 10.93 (SD = 5.32)	M = 10.27 (SD = 5.89)	M = 10.60 (SD = 5.52)
RPQ proactive pre-treatment	M = 6.13 (SD = 4.22)	M = 4.81 (SD = 5.55)	M = 5.45 (SD = 4.91)
RPQ proactive post-treatment	M = 4.07 (SD = 4.18)	M = 3.73 (SD = 3.83)	M = 3.90 (SD = 3.94)
STAS state pre-treatment	M = 12.67 (SD = 6.73)	M = 12.44 (SD = 5.54)	M = 12.55 (SD = 6.04)
STAS state post-treatment	M = 14.73 (SD = 7.18)	M = 12.60 (SD = 5.48)	M = 13.67 (SD = 6.36)
STAS trait pre-treatment	M = 26.33 (SD = 7.81)	M = 24.81 (SD = 8.27)	M = 25.55 (SD = 7.95)
STAS trait post-treatment	M = 24.33 (SD = 8.26)	M = 21.67 (SD = 5.65)	M = 23.00 (SD = 7.09)
BIS pre-treatment	M = 17.33 (SD = 1.84)	M = 16.88 (SD = 1.89)	M = 17.09 (SD = 1.85)
BIS pre-treatment	M = 17.53 (SD = 1.85)	M = 18.40 (SD = 1.99)	M = 17.97 (SD = 1.94)
BAS Reward pre-treatment	M = 16.53 (SD = 1.77)	M = 16.56 (SD = 2.13)	M = 16.55 (SD = 1.93)
BAS Reward post-treatment	M = 16.47 (SD = 2.67)	M = 16.33 (SD = 2.74)	M = 16.40 (SD = 2.66)
BAS Drive pre-treatment	M = 11.53 (SD = 2.53)	M = 11.63 (SD = 3.01)	M = 11.58 (SD = 2.74)
BAS Drive post-treatment	M = 11.13 (SD = 2.72)	M = 10.93 (SD = 4.03)	M = 11.03 (SD = 3.38)
BAS Fun pre-treatment	M = 11.60 (SD = 1.99)	M = 11.25 (SD = 1.98)	M = 11.42 (SD = 1.96)
BAS Fun pre-treatment	M = 11.33 (SD = 1.95)	M = 10.60 (SD = 2.19)	M = 1.97 (SD = 2.08)
Paradigms			
VDT pre-treatment	M = 6.67 (SD = 7.81)	M = 4.19 (SD = 5.31)	M = 5.39 (SD = 6.64)
VDT pre-treatment	M = 6.93 (SD = 8.96)	M = 7.47 (SD = 10.99)	M = 7.20 (SD = 9.86)
HIBT anger pre-treatment	M = 54.52 (SD = 21.01)	M = 60.23 (SD = 24.92)	M = 57.46 (SD = 22.91)
HIBT anger pre-treatment	M = 59.11 (SD = 21.31)	M = 58.48 (SD = 24.85)	M = 58.79 (SD = 22.79)
HIBT disgust pre-treatment	M = 49.79 (SD = 21.49)	M = 51.28 (SD = 32.79)	M = 50.56 (SD = 27.46)
HIBT disgust pre-treatment	M = 56.32 (SD = 23.11)	M = 48.43 (SD = 29.23)	M = 52.24 (SD = 26.28)
HIBT fear pre-treatment	M = 26.45 (SD = 25.92)	M = 24.73 (SD = 31.99)	M = 25.56 (SD = 28.74)
HIBT fear pre-treatment	M = 29.13 (SD = 30.93)	M = 29.02 (SD = 33.38)	M = 29.07 (SD = 31.64)
HIBT happy pre-treatment	M = 12.86 (SD = 14.70)	M = 9.46 (SD = 15.69)	M = 11.10 (SD = 15.07)
HIBT happy pre-treatment	M = 15.35 (SD = 19.96)	M = 9.98 (SD = 12.67)	M = 12.57 (SD = 16.52)
VR-GAIME			
Correctly approached			
Session 1	M = 18.60 (SD = 2.29)	M = 32.13 (SD = 11.84)	-
Session 2	M = 16.60 (SD = 7.53)	M = 34.38 (SD = 8.76)	-
Session 3	M = 17.33 (SD = 6.65)	M = 36.00 (SD = 8.33)	-
Session 4	M = 16.67 (SD = 6.43)	M = 34.69 (SD = 9.77)	-
Session 5	M = 15.86 (SD = 5.16)	M = 36.56 (SD = 8.91)	-
Correctly avoided			
Session 1	M = 17.07 (SD = 5.16)	-	-
Session 2	M = 18.67 (SD = 3.33)	-	-
Session 3	M = 19.40 (SD = 0.63)	-	-
Session 4	M = 18.93 (SD = 1.49)	-	-
Session 5	M = 18.07 (SD = 3.15)	-	-

*Note*: The mean total score on the Social Dysfunction and Aggression Scale (SDAS), the Discrete Emotion Questionnaire, the Self-Report Psychopathy short form (SRP-SF), the Aggression Questionnaire (AQ), the subscales of the Reactive Proactive Questionnaire (RPQ), the state and trait subscale of the State Trait Anger Scale (STAS), the subscales of the Behavioral Inhibition System/Behavioral Activation System (BIS/BAS), the Voodoo Doll Task (VDT), and the Hostile Interpretation Bias Task (HIBT) are reported.

**Table 4 brainsci-11-01484-t004:** Results of linear mixed model.

Model	Parameter	Estimate	95% CI	t	df	*p*
Basic model 1	Intercept	13.41	9.11 – 17.71	6.38	28.99	<0.001
SDAS self-report	Time	−1.97	−4.15 – 0.22	−1.84	29.05	0.076
	Condition	1.19	−4.98 – 7.37	0.39	29.01	0.396
	Time × Condition	0.93	−2.19 – 4.05	0.61	28.63	0.548
Basic model 2	Intercept	10.54	7.34 – 13.73	6.69	34.20	<0.001
SDAS clinician	Time	−1.79	−3.53 – 0.05	−2.12	25.23	0.044
	Condition	3.19	−1.40 – 7.79	1.41	34.39	0.167
	Time × Condition	0.28	−2.25 – 2.81	0.22	26.09	0.825
Basic model 3	Intercept	3.06	2.72 – 3.39	18.68	25.22	<0.001
DEQ	Time	−0.09	−0.31 – 0.13	−0.86	35.98	0.398
	Condition	−0.26	−0.76 – 0.23	−1.09	25.71	0.285
	Time × Condition	0.04	−0.28 – 0.35	0.25	36.14	0.803
Model including main effects baseline characteristics	Intercept	13.08	10.17 – 15.99	10.18	8.87	<0.001
SDAS self-report	Time	−1.49	−2.96 – −0.03	−2.08	30.38	0.046
	Condition	2.72	−1.32 – 6.76	1.48	10.93	0.167
	SRP-SF	−0.19	−0.61 – 0.22	−1.08	12.61	0.324
	AQ	−0.16	−0.48 – 0.15	−1.14	10.56	0.279
	STAS state	0.01	−0.67 – 0.69	0.03	10.11	0.976
	STAS trait	0.56	0.13 – 1.00	2.83	11.14	0.016
	RPQ proactive	0.49	−0.38 – 1.37	1.23	12.69	0.242
	RPQ reactive	0.91	0.13 – 1.70	2.59	10.02	0.027
	BIS	−0.23	−1.29 – 0.83	−0.49	10.26	0.637
	BAS reward	1.12	−0.81 – 3.04	1.28	11.16	0.228
	BAS drive	0.28	−1.06 – 1.62	0.47	9.68	0.649
	BAS fun	−1.55	−3.14 – 0.05	−2.18	9.27	0.056
	VDT	−0.29	−0.66 – 0.09	−1.68	10.43	0.123
	HIBT angry	0.14	−0.04 – 0.33	1.77	8.31	0.114
	HIBT happy	0.19	−0.14 – 0.53	1.27	11.54	0.229
	HIBT fear	0.04	−0.09 – 0.16	0.66	10.49	0.525
	HIBT disgust	−0.12	−0.26 – 0.03	−1.93	6.81	0.097
Model including significant main effects + interaction effects	Intercept	13.22	10.50 – 15.95	9.89	32.31	<0.001
SDAS self-report	Time	−1.28	−2.78 – 0.23	−1.75	24.55	0.093
	Condition	0.69	−2.49 – 3.87	0.45	23.49	0.656
	STAS trait	0.47	0.03 – 0.90	2.19	26.36	0.038
	RPQ reactive	0.87	0.18 – 1.56	2.59	26.36	0.015
	BAS fun	−0.37	−2.08 – 1.34	−0.45	25.77	0.660
	Time × STAS trait	−0.12	−0.43 – 0.19	−0.82	28.14	0.419
	Time × RPQ reactive	0.20	−0.31 – 0.71	0.81	29.51	0.425
	Time × BAS fun	−0.11	−1.31 – 1.13	−0.15	28.24	0.880
	Time × Condition × STAS trait	0.24	−0.18 – 0.65	1.18	24.25	0.250
	Time × Condition × RPQ reactive	−0.55	−1.19 – 0.09	−1.79	23.41	0.087
	Time × Condition × BAS fun	−0.52	−2.17 – 1.13	−0.65	22.05	0.521
Final model	Intercept	13.19	10.55 – 15.85	10.11	34.67	<0.001
SDAS self-report	Time	−1.26	−2.75 – 0.23	−1.73	29.10	0.094
	Condition	0.81	−2.24 – 3.85	0.55	25.41	0.590
	STAS trait	0.40	0.12 – 0.69	2.88	26.59	0.008
	RPQ reactive	0.99	0.38 – 1.61	3.29	33.06	0.002
	BAS fun	−0.61	−1.74 – 0.51	−1.13	25.43	0.270
	Time × RPQ reactive	0.06	−0.38 – 0.51	0.29	36.76	0.774
	Time × Condition × RPQ reactive	−0.46	−0.96 – 0.04	−1.89	26.78	0.070

## Data Availability

The data presented in this study are available on request from the corresponding author. The data are not publicly available due to the privacy and vulnerability of the studied population consisting of forensic psychiatric outpatients.

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
