# Peer review of "Testing the Effects of a Virtual Reality Game for Aggressive Impulse Management: A Preliminary Randomized Controlled Trial among Forensic Psychiatric Outpatients"

_brainsci, 2021, doi:10.3390/brainsci11111484_

Round 1
Reviewer 1 Report
In this article, the authors propose an innovative VR-based intervention for aggression impulses management. According to the results, the authors did not find any significant difference between the experimental and control conditions. I have some comments and doubts which dampen my enthusiasm towards this article, which is really interesting and very well written.
- The authors should provide a definition of approach motivation (line 48)
- In my opinion, the authors should emphasize and justify more the advantages of using VR for their intervention. The authors, for instance, do not mention the efficacy of VR for inducing emotional states (which, in the specific case of an intervention to manage aggressive impulses, seems to be quite relevant), as well as the concept of "sense of presence". I think that a more strong theoretical background regarding the approach used to develop the intervention is needed
- The authors should justify the inclusion and exclusion criteria. For instance, why only male participants? Why did the authors exclude participants with certain disorders? Did the author consider medication? did the authors include also individuals who were already receiving a psychological treatment?
- line 161 "The antisocial and borderline personality disorder as well as the intermittent explosive disorder are the most common psychopathologies": Did the author control for this variable? For instance, differences due to a different diagnosis or severity?
- Details about the VR system used as well as about the development of the environment are missing
- It would be nice to put pictures to compare agreeable and disagreeable avatars
- I think that the article is very well written and that the VR-based intervention developed by the authors is really interesting and potentially innovative. However, my main concern regards the sample. The authors calculated the sample size. Nevertheless, the recruited sample was twice as small. Accordingly, it is not possible to conclude whether the intervention was effective (or not), whether it needs to be improved in some aspects, or whether the lack of results is actually due to statistical issues. This limitation has dampened a little bit my enthusiasm towards this interesting work. I think it would be worthy to try to recruit more individuals to be integrated in this sample and reach the needed sample size. It would be useful to disentangle and understand the efficacy of the treatment which, as aforementioned, is really innovative.
- line 577 "Moreover, it may be that forensic psychiatric outpatients are not uniformly characterized by reactive approach motivation." I totally agree, it would have been useful to somehow investigate this aspect among participants and control it in the analyses.
- line 588 "As a results, it is possible that the final version of the VR-GAIME diverged so much from the original joystick training [21] that it could no longer be expected to have the same effects." Again, I agree with this statement. Consistently, it would have been useful to first compare the original game and the VR based game to check whether the findings are consistent (in other words, to somehow validate the new VR game)
- line 600: "Even though the VR-GAIME did not yield the expected reduction of aggressive behavior during treatment, the game did have another important clinical effect. Several patients reported that playing the game gave them insight in their own as well as in others behavior". On which basis do the authors reach this conclusion? a focus group or qualitative interview?
- What about the future directions? Which is the next step to improve this intervention?
TYPOS - The manuscript needs to be revised for typos (see for example line 53 "the latter findings suggests..."; line 84 "Serious games refer to games, even though they are fun and engaging, have training, education or health improvement are their primary purpose" something is wrong in this sentence)
Author Response
Dear reviewer,
We revised the manuscript according to your very helpful suggestions. We have listed our responses in itemized fashion. We believe that the manuscript has improved thanks to the comments and suggestions, and hope that you agree with us that the revised version is suitable for publication in Brain Sciences.

Reviewer 2 Report
The Authors report on the testing of effects of a newly developed serious game training - rooted in the motivational approach to anger management - as an additional treatment option to conventional aggression regulation interventions. The RCT is well conducted and clearly reported on a topic of great significance – aggression behavior regulation. I commend the Authors on this endeavor, and, in particular, for the inclusion of forensic psychiatric outpatients. The discussion is enriched by a thorough analysis of potential explanations for lack of expected effects.
The major remark I have for the Authors' consideration, which I hope will help to further strengthen the paper and its contributions, regards the “translation” of the intervention into a gamified VR-format. As the Authors declare, many design decisions were taken in absence of relevant literature on the specific domain, and without a clear theoretical rationale. Yet, these decisions were actually taken.
At present, in the literature on innovative simulation technologies for clinical applications, the need to clarify design options and to identify which specific ones impact on specific outcomes is acknowledged as of great importance to prevent the blurring of the key moderators/mediators of a clinical intervention. So, focusing on “how” design choices impact outcomes could really boost the potential to inform more efficient and effective technology-driven simulation interventions for anger regulation. Along this line, stating the theoretical and clinical rationale of the key design choices in developing the VR-GAIME could benefit this relevant debate.
Another remark regards the articulated theoretical framework connecting anger, anger regulation, aggressive behavior, aggression regulation problems. Throughout the paper, it does not appear crystal clear to me.
Just few examples: anger definition (line 26), experience of anger and disposition to experience it (trait anger) (line 28), aggressive behavior (lines 30-32), aggression regulation problems/aggression regulation (lines 35, 46, 69, 629), anger and aggression regulation problems (79), managing anger and aggression (line 633). As Authors also acknowledge, anger regulation and aggression behavior regulation do not stem from the same theoretical frameworks. In the case of the emotion of anger, for instance, the “process model” of emotion regulation suggests that strategies can be attempted at any stage of the emotion generation process (e.g., situation selection, situation modification, attentional deployment, cognitive change, and response modulation; Gross, 2015) to modulate an emotion or potential occurrence of emotion. And strategies (including avoidance) can have both cognitive and behavioral components. So, for instance, what does the expression “aggression regulation problems” stand for? Are they instances of emotion dysregulation? And, if so, how do they match with the type and target of training developed? And how was the specific assumption operationalized into the gamified VR-training?
Indeed, not only there is some debate on what can be considered as an emotion regulation strategy, but the incorporation of findings from contemporary emotion regulation research into aggression research is still on the way.
Nonetheless, I believe the paper would benefit from the disentangling of Authors’ theoretical assumptions on anger, aggression, and anger regulation, and aggression regulation, at least in the Introduction, and, possibly, in unveiling the rationale of key design options of the VR-GAIME.
A third minor remark still has to do with the “translation” of the avoidance training into a gamified experience. The Authors state that “to warrant sufficient treatment motivation, it would be desirable to develop a more engaging variant of the motivational training” (lines 80-82). But, besides the assumption that serious games can enhance motivation, was engagement in the training measured throughout the interventions – and how? Given that both the TAU and the VR-GAIME interventions “occurred either in 229 groups (N = 24) or individually (N = 7) and consisted of two 90-minutes weekly sessions during 12 weeks" (lines 229-231), data on engagement – of course depending on the type of measure – could provide insights on how the simulated experience was appraised by participants.
Author Response

(The authors gave the same response as above.)

Round 2
Reviewer 1 Report
Thank you very much to the authors for considering my suggestions. My main concern is still the sample size. I completely understand the authors' point, it is often very hard to recruit big samples (especially with no financial resources). However, recruiting the right sample size is essential in order to reach reliable conclusions. I consider this issue to be the major problem of the paper from a methodological point of view, as the study is extremely underpowered (twice as small).
I would reccomend the authors to at least change the title in order to emphasize that this is a preliminary study with preliminary results. Another important issue is the lack of validation of the VR-based game. The authors translated this VR game from the original joystick version, but no validation was performed. We cannot exclude that the two versions are divergent and that the lack of results are partially the results of this limitation.
To conclude, I just have a few minor comments
- exclusion criteria: please, specify how mental disorders were assessed (interview? MINI?)
- please, include male sex as a limitation of the study. Even if the majority of your patients are male, there are still female patients who could benefit from the intervention and who were not included in this study
Author Response
Thank you for reviewing our manuscript. We believe that the manuscript has improved again thanks to your comments and suggestions, and hope that you agree with us that the revised version is suitable for publication in the special issue “Dimensions of Pathological Aggression: From Neurobiology to Therapy” of Brain Sciences.

Reviewer 2 Report
I agree with the update of the manuscript. It sounds clearer and more focused. Thank you.
Author Response
Thank you for reviewing our manuscript.